# Metaprofiling of the Bacterial Community in Colonized Compost Extracts by *Agaricus subrufescens*

**DOI:** 10.3390/jof8100995

**Published:** 2022-09-22

**Authors:** Matheus Rodrigo Iossi, Isabela Arruda Palú, Douglas Moraes Soares, Wagner G. Vieira, Lucas Silva Alves, Cassius V. Stevani, Cinthia E. C. Caitano, Samir V. F. Atum, Renato S. Freire, Eustáquio S. Dias, Diego Cunha Zied

**Affiliations:** 1Programa de Pós-Graduação em Microbiologia Agropecuária, Faculdade de Ciências Agrárias e Veterinárias (FCAV), Universidade Estadual Paulista (UNESP), São Paulo 14884-900, Brazil; 2Faculdade de Ciências Agrárias e Tecnológicas (FCAT), Universidade Estadual Paulista (UNESP), São Paulo 17900-000, Brazil; 3Departamento de Química Fundamental, Instituto de Química, Universidade de São Paulo (USP), São Paulo 05508-220, Brazil; 4Departamento de Bioquímica, Instituto de Química, Universidade de São Paulo, São Paulo 05508-220, Brazil; 5Departamento de Biologia, Universidade Federal de Lavras (UFLA), Lavras 37200-900, Brazil

**Keywords:** *Agaricus blazei*, metagenomics, microbiomics, mushroom production, 16S rDNA, nanopore sequencing

## Abstract

It is well-known that bacteria and fungi play important roles in the relationships between mycelium growth and the formation of fruiting bodies. The sun mushroom, *Agaricus subrufescens*, was discovered in Brazil ca. 1960 and it has become known worldwide due to its medicinal and nutritional properties. This work evaluated the bacterial community present in mushroom-colonized compost extract (MCCE) prepared from cultivation of *A. subrufescens*, its dynamics with two different soaking times and the influence of the application of those extracts on the casing layer of a new compost block for *A. subrufescens* cultivation. MCCEs were prepared through initial submersion of the colonized compost for 1 h or 24 h in water followed by application on casing under semi-controlled conditions. Full-length 16S rRNA genes of 1 h and 24 h soaked MCCE were amplified and sequenced using nanopore technology. Proteobacteria, followed by Firmicutes and Planctomycetes, were found to be the most abundant phyla in both the 1 h and 24 h soaked MCCE. A total of 275 different bacterial species were classified from 1 h soaked MCCE samples and 166 species from 24 h soaked MCCE, indicating a decrease in the bacterial diversity with longer soaking time during the preparation of MCCE. The application of 24 h soaked MCCE provided increases of 25% in biological efficiency, 16% in precociousness, 53% in the number of mushrooms and 40% in mushroom weight compared to control. Further investigation is required to determine strategies to enhance the yield and quality of the agronomic traits in commercial mushroom cultivation.

## 1. Introduction

*Agaricus subrufescens* is a Brazilian Agaricales fungus, the mushrooms of which are considered functional foods with medicinal and nutraceutical properties, resulting in its popularization across the world in recent decades [1]. As mushroom cultivation does not require special conditions, this species is cultivated in a rustic way without controlled conditions (e.g., temperature, relative humidity, light period) by small growers, contributing to increases in family incomes [2,3,4]. In fact, this type of cultivation can, in some cases, favor productivity when linked to the adaptability of the strain.

Mushrooms of the genus Agaricus require a casing, a peat layer over the compost colonized by the mycelium, to fructify [5]. Aside from providing physical support for the formation of mushrooms [6], this creates the physicochemical and biological conditions that promote fruiting [7]. The material used must meet some requirements, including a medium that is close to neutral (pH 7), high water-retention and -release capacities, porosity that allows the exchange of gases between the substrate and the medium and resistance to stabilization after frequent irrigation [6,7,8,9,10].

Despite the existence of some reports on the casing irrigation in *Agaricus bisporus* cultivation [9,11], there are few specific studies for *A. subrufescens*, which opens the possibility of exploring this technique with the aim of improving production performance. Furthermore, the concomitant use of chemical and/or biological agents with the irrigation can provide an opportunity to modify the growth medium in the casing layer and contribute to increasing the yield and quality of mushrooms, as successfully demonstrated by the application of fungicides at low concentrations [12] and by a study that used a spent mushroom substrate extract as a biological control for mushroom diseases [13].

In addition to the physicochemical parameters, bacteria–fungi interactions also have a strong influence on mushroom production, ranging from antagonism to mutualism. Recent reports have attributed an important role to the compost and casing microbiome in the growth and fructification of cultivated mushrooms, mainly by influencing the quality and maturation of the compost, mycelium running, primordia induction and formation of fruit bodies [5,14,15].

In addition to providing water supply and maintaining a humid microclimate for the casing, the use of mushroom-colonized compost extract (MCCE) may modify the microbiome of the casing. Thus, the characterization of microbial communities occurring in MCCEs and the effects of their application on the cultivation of commercial mushrooms may reveal important bacteria–fungi associations related to higher yields and superior quality for agronomic traits [5]. Current long-read DNA sequencing technology, including portable nanopore devices, have been successfully applied for the analysis of full-length 16S rRNA gene amplicons, providing a reliable, rapid and cost-effective strategy to assess bacterial communities [16,17,18].

In this scenario, we evaluated the bacterial diversity of MCCE obtained from the cultivation of *A. subrufescens* and the effects of its application on the casing layer of a new compost block for the mushroom’s cultivation. This is the first report on 16S rRNA gene-based nanopore metagenomics applied to the microbiome analysis of substrates used for *A. subrufescens* production. Together with data from agronomic traits of mushrooms, these new findings may help to bring new insights about the influences of the bacterial communities, occurring in both the compost and casing on *A. subrufescens* mycelium growth.

## 2. Materials and Methods

### 2.1. Experimental Design

Experiments were conducted under a completely randomized design and carried out in a double factorial scheme (5 application dates × 3 extraction times, including control) including 15 treatments with 4 replications, resulting in 60 experimental units.

### 2.2. Compost Preparation

The substrate was prepared using the traditional method of composting for the cultivation of *A. bisporus*. Briefly, phase 1 lasted 22 days and phase 2 lasted 7 days, totaling 29 days. For the substrate formulation, 900 kg of wheat straw, 1400 kg of sugarcane bagasse, 45 kg of soybean meal, 4.5 kg of urea, 5 kg of ammonium sulfate, 10 kg of simple superphosphate and 40 kg of limestone were used. Bulky materials (wheat straw and sugarcane bagasse) were moistened for 9 days and turned over every 2 days. During phase 1, the concentrated materials (soybean meal, urea, ammonium sulfate, simple superphosphate and limestone) were added over the substrate windrow after each turn [19]. In phase 2, the substrate was maintained for 18 h at 59 ± 1 °C for pasteurization, followed by conditioning for 5 days at 47 ± 2 °C. All materials were purchased from Compobras^®^ (Castro, PR, Brazil), and the mineral analysis is shown in Appendix A.

### 2.3. Spawn Production

The strain used in the experiments was *A. subrufescens* ABL 04/49, which is deposited in the public mycological collection of the FCAT at the Centro de Estudos em Cogumelos (Dracena, SP, Brazil). The inoculum was produced from sorghum grains (*Sorghum bicolor*) cooked for approximately 40 min in boiling water. After cooking, excess water was drained and 2% calcite was added to the grains based on their wet weight to provide calcium and raise the pH [20]. After homogenization with calcite, the grains were autoclaved for a period of 4 h at 121 °C. After cooling, the grains were inoculated under aseptic conditions in a laminar flow chamber. The inoculum was incubated at a temperature of 28 °C. Substrate inoculation was performed using 1% inoculum for each experimental unit, which contained 2.8 kg wet weight. The experiments were conducted in a semi-controlled chamber with only a cooling system, which resulted in thermal oscillations between day and night periods.

### 2.4. Casing Layer and Fruiting Induction

Peat moss (+terra^®^, Castro, PR, Brazil) was used as casing layer, deposited approx. 3.0 cm above the leveled colonized compost [3]. The ruffling was performed before the mycelium reached the surface of the casing layer, 6 days after the addition of the casing [21]. After 24 h of ruffling, when the mycelium resumed its metabolic activity, the fruiting induction process began, and the temperature was decreased by 2 °C per day until it reached 20 °C, then increased by 2 °C per day until it reached 26 °C [21]. In the harvest phase, the CO_2_ content ranged from 600 to 1000 ppm.

### 2.5. Mushroom-Colonized Compost Extract (MCCE) Preparation and Addition to the Casing Layer

For the preparation of the extract, the compost colonized with the *A. subrufescens* ABL 04/49 strain 16 days after spawning was submerged in tap water in a 1:5 (substrate: water) ratio (Figure 1). Two MCCEs were prepared with different soaking times: 1 h and 24 h. Subsequently, the substrate was removed and the solution was filtered through a 10 mesh (2.0 mm) sieve. MCCEs were produced from the dilution of the filtered solution to a concentration of 12.5%; i.e., 87.5 L of water was added to 12.5 L of compost extract. Each experimental unit—boxes 15 cm × 30 cm × 40 cm (h × w × l) containing 2.8 kg of compost—was watered with 100 mL of either 1 h or 24 h soaked MCCE on the 3rd, 6th, 9th, 12th and 15th day after the addition of the casing layer to the compost. This volume was equivalent to 2.212 L of irrigation per square meter. The control treatment consisted of the water irrigation over the casing layer without MCCE.

### 2.6. Harvest Phase

The fruiting period started 17 days after the addition of the casing layer. Harvesting was performed manually up to three times a day and mushrooms were harvested with the closed cap [22]. Post-harvest procedures included the cutting of the base of the stipe and the washing of mushrooms to clean the cap and remove casing layer residues.

### 2.7. Agronomic Trait Evaluation

Data from agronomic traits collected daily were used to estimate the biological efficiency, precociousness, number and weight of the mushrooms. All parameters were calculated following Atila [23]. Biological efficiency was expressed as the ratio of the fresh mass of mushrooms and the dry mass of the initial substrate. Precociousness was calculated as the production of the first half of the harvest time divided by the total production. The number of mushrooms was calculated as the sum of all collected basidiomes. The weight of mushrooms was obtained from the ratio of the total mass of fresh mushrooms to the number of basidiomes collected.

### 2.8. Statiscal Analysis

Agronomic traits were measured in four replicates for the control and the 1 h and 24 h soaked MCCEs for each application date (Appendix A). Standard deviations were calculated from results obtained under different conditions, with the presentation data of the media, accompanied by the range of variation verified in each repetition within the treatment using SAS statistical software (SAS Institute Inc., Cary, NC, USA) (Appendix A).

### 2.9. Estimation of Bacterial Diversity by 16S rRNA Gene Sequencing

In order to characterize the 16S rRNA gene-based microbiome, 1 h and 24 h soaked MCCEs were filtered and the filter surface was used for DNA isolation with the DNeasy Power Water kit (QIAGEN, Germany), according to the manufacturer’s instructions. PCR amplification of full-length 16S rRNA genes was carried out in 50 µL reactions using Taq DNA Polymerase (Cellco Biotech, Sao Carlos, Brazil) and the universal primers 27F and 1492R [24]. PCR reactions were incubated in the SimpliAmp thermal cycler (Applied Biosystems, Waltham, MA, USA) under the following settings: initial denaturation at 95 °C for 2 min, 35 cycles of denaturation at 95 °C for 30 s, annealing at 52 °C for 30 s, extension at 72 °C for 90 s and a final extension at 72 °C for 120 s. DNA amplicons (~1.5 kb) were separated by agarose gel electrophoresis and purified using the QIAquick PCR purification kit (QIAGEN, Hilden, Germany). DNA libraries were prepared from 2 µg of the purified DNA amplicons. Double-strand DNA molecules were end-repaired using the NEBNext FFPE DNA Repair Mix and NEBNext Ultra II End Repair/dA-tailing Module reagents (New England Biolabs, Ipswich, MA, USA). A total of 10 unique DNA barcodes from the Native Barcoding Expansion kit (EXP-NBD104) (Oxford Nanopore Technologies, Oxford, UK) were attached to the end-prepped DNA (1–5 for 1 h and 6–10 for 24 h soaked MCCE samples). Equimolar amounts (350 ng) of each barcoded sample were combined to produce a pooled sample, used for the adapter ligation with the Ligation Sequencing kit (SQK-LSK109) (Oxford Nanopore Technologies, UK). The DNA library was cleaned up using the Agencourt AMPure XP beads (Beckman Coulter, Brea, CA, USA) and quantified using the Qubit fluorometer (Applied Biosystems, USA). A sequencing library was loaded into the minION Mk 1B flow cell R9.4.1 and the nanopore sequencing was performed over 48 h.

### 2.10. Bioinformatic Analysis

Basecalling, demultiplexing and quality control were undertaken in guppy v. 5.0.16 with the high quality r9.4.1_450bps_hac model. Screening against possible contaminants was conducted using BLAST v. 2.12 [25] against the UniVec database v. 10.0 (http://www.ncbi.nlm.nih.gov/tools/vecscreen/univec/, accessed on 14 August 2022). Nanofilt v. 2.8.0 [26] was used to filter the reads by size, selecting only those between 1400 and 1600 bp. After filtering, kraken2 v.2.1.1 [27] was used with the Greengenes database (v. 13.5 downloaded at 24 March 2022) for classification. To support the Greengenes results, minimap2 v. 2.22 [28] was also used to align the reads to the NCBI 16S RefSeq database (https://www.ncbi.nlm.nih.gov/refseq/targetedloci/16S_process/—downloaded on 24 March 2022). For visualization of the results, Pavian v. 1.0 [29] was used to generate Sankey graphs and classification tables of phyla and genera ranked by the number of reads. Data from barcode A, corresponding to the 16S-based microbiome for 1 h soaked MCCE, were combined, as were those from barcode B, representing the 24 h soaked MCCE samples (Appendix A).

## 3. Results 

### 3.1. Composition and Dynamics of MCCE Microbiome

To investigate the bacterial diversity and the microbiome dynamics in 1 h and 24 h soaked MCCEs, these solutions were initially filtered, and the membrane was used for DNA isolation. The full-length 16S rRNA gene was successfully amplified, and the amplicons were properly barcoded and prepared for nanopore sequencing. After bioinformatics processing, including basecalling, demultiplexing and quality control steps, 1,269,157 reads were obtained and 99.85% of them were taxonomically classified.

Proteobacteria, followed by Firmicutes and Planctomycetes, were found to be the most abundant phyla in both 1 h and 24 h soaked MCCE (Figure 2 and Figure 3A). Proteobacteria was the dominant phylum in both MCCEs and its abundance was noticeably increased from 53.8% in 1 h soaked MCCE to 97.1% in 24 h soaked MCCE, which suggests that these changes in microbiome dynamics were influenced by the duration of the soaking step during MCCE preparation. Most Proteobacteria were assigned to the Gammaproteobacter class, which includes several Gram-negative microbes with scientific and health relevance, such as the families Enterobacteriaceae, Vibrionaceae and Pseudomonadaceae.

More than 92% of Firmicutes, the second most abundant phylum, were classified as Bacilli, a Gram-positive, aerobic and spore-forming bacteria class. The abundance of Firmicutes was dramatically reduced from 23% in 1 h soaked MCCE to 1.7% in 24 h soaked MCCE, which may have been related to lower oxygen availability during the longer soaking times, causing a reduction in the growth rate of this class of microorganisms. 

*Serratia* was the most abundant genus in both MCCEs, increasing from 19.3% in 1 h soaked MCCE to 39.7% in 24 h soaked MCCE (Figure 3B). Likewise, the abundance of *Klebsiella* genus was slightly increased in 24 h soaked MCCE (from 7.2% to 10.3%). In addition to these two common genera, *Bacillus*, *Paenibacillus* and *Planctomyces* were the most abundant in 1 h soaked MCCE and *Pseudomonas*, *Citrobacter* and *Erwinia* in 24 h soaked MCCE (Figure 3B).

A total of 275 different bacterial species were classified from the 1 h soaked MCCE samples (Appendix A) and 166 species from the 24 h soaked MCCE (Appendix A), indicating a reduction in the bacterial diversity with longer soaking times during the preparation of MCCE. In both groups of samples, *Serratia marcescens* was the dominant species, with abundance significantly higher in 24 h soaked MCCE when compared to that obtained after 1 h of soaking (Appendix A). In a more discreet way, the abundance of *Klebsiella oxytoca* was positively correlated to *S. marcescens*, being slightly higher in 24 h soaked MCCE (Appendix A). A longer soaking time also contributed to increasing the abundance of bacteria from the *Pseudomonas* genus, which accounted for up to 15.9% of the total classified genera in 24 h soaked MCCE samples (Figure 3B).

A strong reduction in the percentage of genera classified as “others” was observed in the microbiome of the 24 h soaked MCCE samples (25.9%) in comparison to the 1 h soaked MCCE (56.7%), as well as a significant increment in the representativeness of the top five most abundant genera in the 24 h soaked MCCE microbiome (corresponding to 74.1% of the total genera classified) compared to the 1 h soaked MCCE microbiome (representing only 43.3% of the total genera), indicating a scenario of dramatic changes in the dynamics of the microbial diversity, and its reduction in MCCE prepared with a longer soaking time (Figure 3B).

### 3.2. Evaluation of the Application of MCCE on the Agronomic Traits of Mushrooms

Longer soaking time also contributed to improving all agronomic parameters. Figure 4 presents the average of the application times (3rd, 6th, 9th, 12th and 15th days after casing) with the standard deviation for each analyzed variable. 24 h soaking of the MCCE increased the biological efficiency by 25%, the precociousness by 16%, the number of mushrooms by 53% and the weight by 40% compared to control (Appendix A).

Regarding the 1 h soaked MCCE, it stood out for its shorter preparation time and lower precociousness. Despite the fact that the biological efficiency did not increase significantly, the precociousness decreased, which allowed a shorter crop cycle and a substantial increase of 46% in the weight of mushrooms. Evaluating the best duration for the application of MCCE in the casing layer, and considering the significant increase in biological efficiency, it is recommended to apply the 1 h MCCE between the 9th and 12th days and the 24 h MCCE on the 12th day after casing (Figure 5). The red band indicates the value of the control treatment, with biological efficiency of 20 ± 5%. The results obtained for each agronomic parameter are described in Appendix A.

## 4. Discussion

Mushroom production technology has advanced with the optimization of agronomic parameters related to chemical, physical-chemical, microbiological and environmental aspects [30]. Several studies have demonstrated the role of microorganisms in mushroom production [31,32,33], and analysis of the microbiome in basidiomes revealed that it is highly influenced by the bacterial diversity of the casing [34]. It is evident that, from the inoculation phase to the fructification and harvesting of the mushroom, there is a microbial succession in the compost and casing. Actually, there are several reports concerning the changes in the microbiome dynamics in the casing and compost during mushroom cultivation [34,35,36].

Compared to the classical approaches used to assess microbial diversity, next-generation sequencing (NGS) offers a more powerful and holistic tool for the taxonomic classification of microorganisms, and it is able to detect them even in complex environmental samples and does not require time-consuming steps for enrichment, isolation and cultivation. Most of metagenomics studies characterizing microbiomes from substrates for mushroom cultivation rely on DNA sequencing of the V3-V4 portion of the 16S rRNA gene, a hypervariable region commonly used for bacterial taxonomy, using the short-read technology of Illumina platforms [34,36]. In contrast, we present here the first successful report of a full length 16S rRNA gene-based metagenomics analysis of *A. subrufescens* mushroom-colonized compost extracts (MCCEs) using the long-read sequencing technology of a portable nanopore sequencer (minION, Oxford Nanopore Technologies, UK), providing more complete genetic data for a proper taxonomic classification of reads.

Our sequencing data revealed the Proteobacteria to be the dominant phylum, with a significant increment in 24 h soaked MCCE. This finding is in accordance with previous microbiome analysis in the casing throughout the crop cycle of *A. bisporus*, a mushroom-forming species of the same genus of *A. subrufescens* [34,37]. Other abundant phyla, including Actinobacteria, Bacteroides, Firmicutes, Proteobacteria and Planctomycetes, were also found during the *A. bisporus* mushroom cultivation [15,34,38], and may be related to the beneficial relationship between these bacterial phyla and *Agaricus* fungi. 

Similarly, Firmicutes and Actinobacteria were also found to be dominant phyla in the microbiomes of different developmental stages during *Hypsizygus marmoreus* mushroom cultivation [36]. Firmicutes bacteria mostly belong to the class Bacilli, represented by the most abundant genera *Bacillus* and *Paenibacillus*. Several studies have reported the use of *Bacillus* strains, naturally occurring in casings, as biological agents to suppress the growth of fungal pathogens and increase the yield of commercial mushrooms [5,39,40]. Despite being the third most abundant genus in the microbiome of the 1 h soaked MCCE, *Bacillus* abundance was strongly reduced with longer soaking time, being replaced with *Pseudomonas* in 24 h soaked MCCE.

Likewise, the relative abundance of *Pseudomonas* in the casing increased during the crop cycle of *A. bisporus*, becoming the second most abundant genus by the end of the second flush [34]. The beneficial effects of the proliferation of this genus in the casing are well-documented and mainly involve assisting the hyphae in decomposing complex substances present in the substrate and facilitating the efficient absorption of nutrients [36]. In addition, some species, such as *Pseudomonas putida* can break down the 1-octen-3-ol bond and induce fruiting body formation in *A. bisporus* [41,42]. Similarly, *Pseudomonas fluorescens* is able to remove *A. bisporus* deposits of crystalline calcium oxalate, a prerequisite for fruiting initiation [43], as well as promote the formation of the primordium and enhance the development of fruiting bodies of *Agaricus bitorquis* [42], *Pleurotus eryngii* [44] and *Pleurotus ostreatus* [45].

In both MCCE microbiomes, the Gram-negative and facultative anaerobic bacteria *K. oxytoca* and *S. marcescens* were found, *S. marcescens* being the most abundant bacterial species. It is known that *S. marcescens* exhibits high efficiency in the degradation of chitin through the activity of the chitinase enzyme [46]. Fungi have chitin in the cell wall [47], which contributes to the development of *S. marcescens* in this environment. The dominance of the *Serratia* genus was also reported during the mature period of *H. marmoreus* cultivation. In fact, there are several reports on the benefits of *Serratia* on mushroom cultivation, including its role in biocontrol [48,49], phosphorus solubilization [50] and growth promotion [36].

A recent report showed that the *Serratia odorifera* HZSO-1 strain, a symbiotic bacterium that resides in the *H. marmoreus* hyphae, can shorten the fruiting cycle by 3–4 days and increase the fruiting body yield by 12% [36]. Among secondary metabolites produced by *S. odorifera* HZSO-1, the secretion of quorum-sensing molecules, such as N-acyl homoserine lactones, could stimulate *H. marmoreus* to secrete some lignin-degrading enzymes, promoting growth and development and increasing the yield of mushrooms [36,51]. Furthermore, it has been demonstrated that the *Serratia plymuthica* PRI-2C strain produces the metabollite sodorifen in response to volatile organic compounds (VOCs) emitted by the fungal pathogen *Fusarium culmorum*, suggesting that terpenes may be important molecules in the communication between fungi and bacteria [52]. In this context, it seems plausible that the application of *S. marcescens*-containing MCCE could be explored towards obtaining better agronomic traits in mushroom-forming fungi.

The optimal date for application of MCCE to the casing layer for maximal biological efficiency was found to be the 12th day, when the formation of clumping occurred before the pin phases. During the first week after casing, there was a large increase in the bacterial population, which tended to stabilize or decrease after the 10th day. Precautions should be taken regarding the use of MCCE, such as using the same strain/species of the fungus as the one to be cultivated, checking for the absence of pests and diseases in the colonized compost and, finally, using newly colonized compost with high mycelial growth, which denotes an active metabolic state. The precautions should be the same as for the “compost add casing—CAC’ing” technique, described in detail by Pardo-Giménez et al. [53]. Thus, the use of mushroom growth-promoting microorganisms can involve two technological application levels: one with high investment, involving specific culture medium (solid or liquid) and known and improved species for such application; and another with low investment, involving a simple extraction process (known as “on farn”), with the selection of natural species that represent a high diversity of microorganisms.

Overall, the recent report on *H. marmoreus* suggests that the cultivation of edible mushrooms by adding beneficial microbes to the substrate is a feasible practice [36]. Several microorganisms present in these environments may strongly influence, and in some cases, are required for, the growth and fructification of cultivated mushrooms [5]. Understanding the dynamics of the microbiome in the compost and casing may determine the composition of the mushroom holobiont, the association of fungus and microorganisms [5]. Therefore, the elucidation of the microbiota structure and the symbiotic bacterial communities in mushroom substrates is a key step that can be explored for isolation of growth-promoting strains related to higher yield and better agronomic traits in mushrooms and for the development of synthetic communities for application to mushroom production and disease control [34,36].

## 5. Conclusions

Metagenomics of extracts prepared from *A. subrufescens* compost with different soaking times were used as a powerful tool to elucidate the dynamics of bacterial diversity and bacterial interactions with the hyphae of basidiomycetes during its cultivation. In this context, the long-read technology of nanopore sequencing is a great choice for full-length 16S rRNA gene-based metagenomics, providing the genetic data to explore the taxonomic classification in microbiome analysis. The 24 h soaked MCCE increased the biological efficiency, precociousness, number and weight of the mushrooms. Further investigation is required to determine strategies to enhance the yield and quality of agronomic traits in commercial mushroom cultivation. 

## Figures and Tables

**Figure 1 jof-08-00995-f001:**
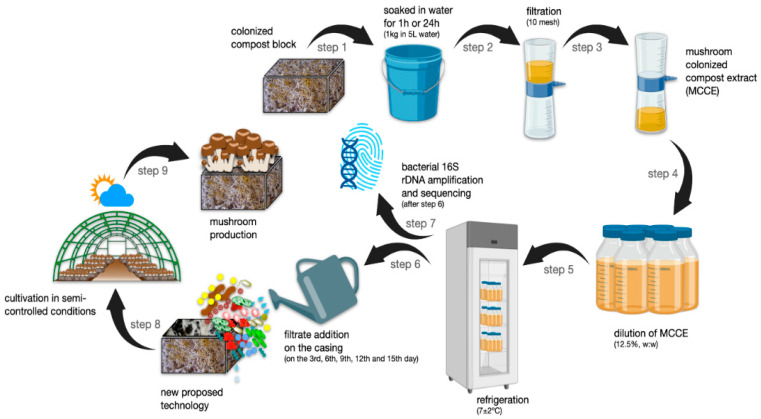
Steps involved in the preparation of mushroom-colonized compost extract (MCCE) and its addition over the casing of compost blocks for the production of *Agaricus subrufescens*. The fully colonized compost block with the mycelium was ground by hand and soaked in water for 1 h or 24 h (step 1), followed by filtration (step 2), dilution and storage of the MCCE in the refrigerator (steps 3 to 5). Each bottle obtained with 1 h or 24 h soaking time was then added to another colonized compost block in the casing on the 3rd, 6th, 9th and 15th days after casing (step 6). Full-length 16S rRNA genes of the 1 h and 24 h soaked MCCEs were amplified and sequenced using nanopore technology (step 7). *A. subrufescens* mushrooms were grown in a chamber under semi-controlled conditions (steps 8 and 9).

**Figure 2 jof-08-00995-f002:**
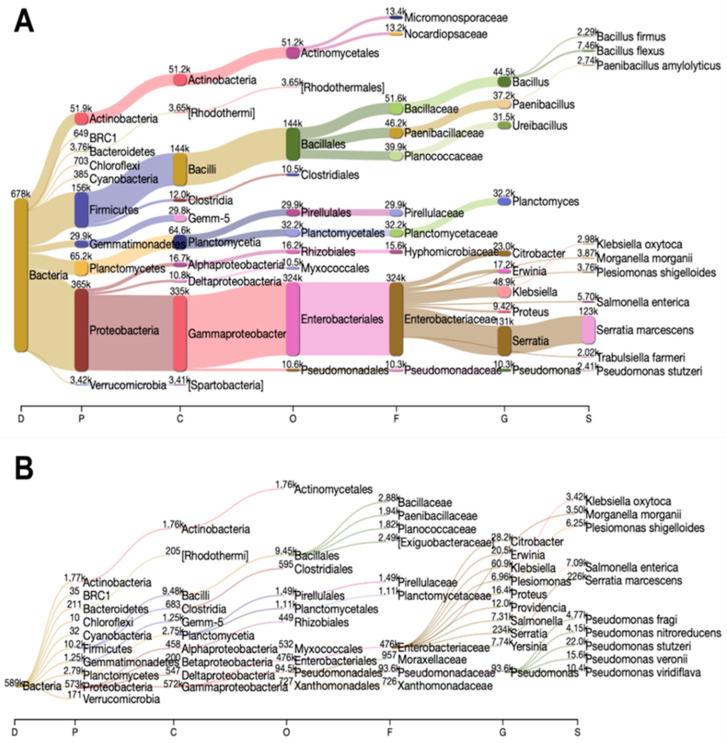
Sankey visualization of taxonomic classification at domain (D), phylum (P), class (C), order (O), family (F), genus (G) and species (S) levels based on the 16S rRNA gene amplicons from samples of mushroom-colonized compost extracts (MCCEs) soaked in water for either 1 h (**A**) or 24 h (**B**).

**Figure 3 jof-08-00995-f003:**
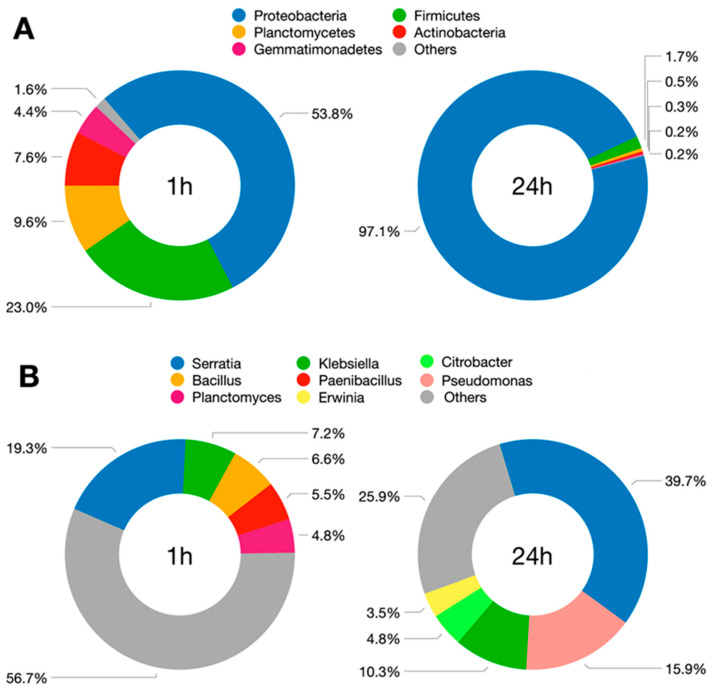
Most abundant phyla (**A**) and genera (**B**) of bacteria present in the *Agaricus subrufescens* mushroom-colonized compost extracts (MCCE) soaked in water for either 1 h or 24 h. Others figures refer to Bacteroidetes, BRC 1, Chloroflexi, Cyanocateria, OD 1 (genera) and Verrucomicrobia (phyla).

**Figure 4 jof-08-00995-f004:**
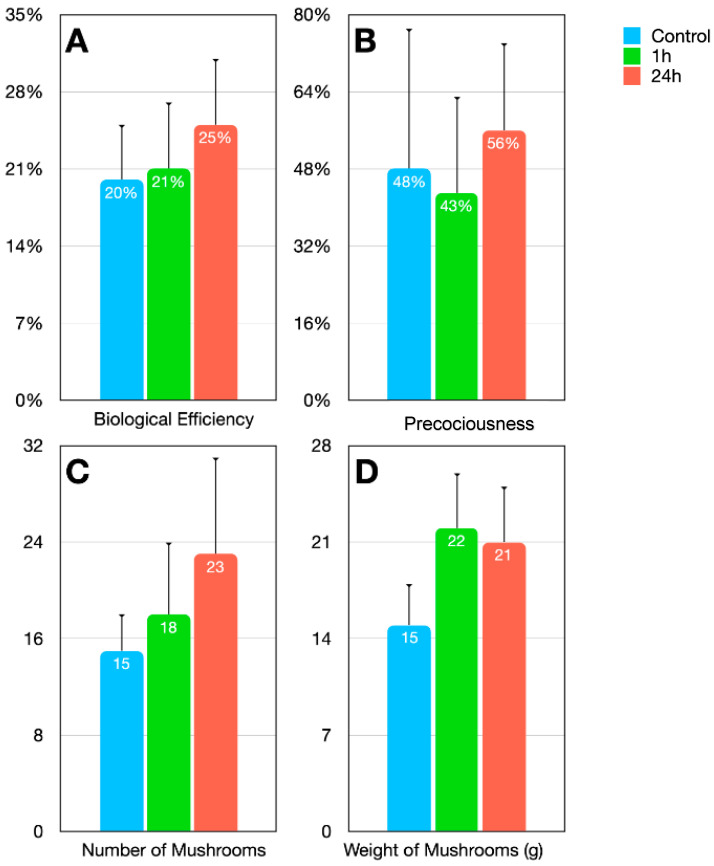
Agronomic parameters of mushrooms cultivated with the control and application of 1 h- and 24 h soaked MCCE over the casing, which (**A**) indicate biological efficiency, (**B**) precociousness, (**C**) number of mushroom, and (**D**) weight of mushroom.

**Figure 5 jof-08-00995-f005:**
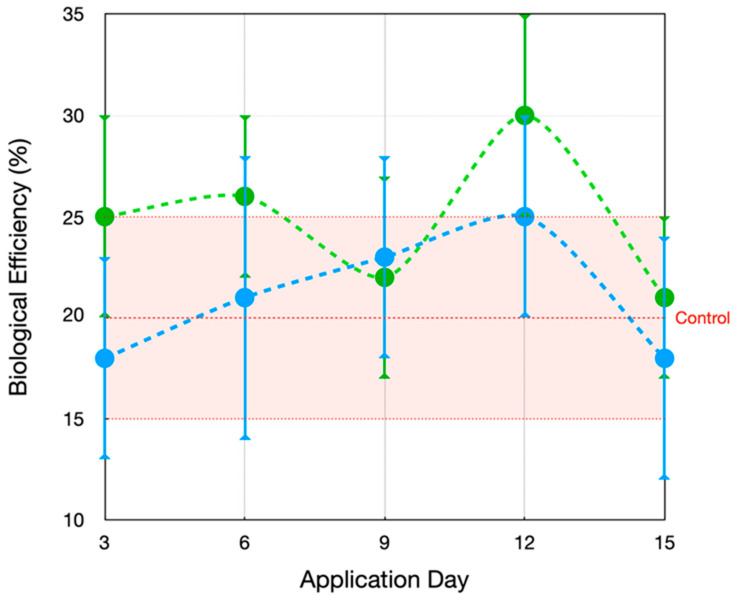
Biological efficiency of the *Agaricus subrufescens* mushroom cultivation after the application of 1 h or 24 h soaked MCCE on the 3rd, 6th, 9th, 12th and 15th days after casing. The red line corresponds to the control treatment. The highlighted red area includes the control mean value and standard deviation (20 ± 5%, *n* = 4).

## Data Availability

Not applicable.

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
