# Peer review of "Metaprofiling of the Bacterial Community in Colonized Compost Extracts by Agaricus subrufescens"

_jof, 2022, doi:10.3390/jof8100995_

Round 1
Reviewer 1 Report
Corrections needed in the manuscript:
Line# 143: misspelled "Plhase"
L#150: mushroom 'precocity' is unlikely to be understood by most readers even with the attempted description. Is there another term that could be used.
L#216: change 'Over' to 'More"
L#291: change 'on' to 'in'
L#296: change 'requiring' to 'require'
L#297: delete 'to'; substitute 'characterizing'
L#303: more complete than what?
L#316: italicize Bacillus
L#317: change 'fungi' to 'fungal'
L#322: delete 'it'
L#323: change 'benefic' to 'beneficial'
L#325: ad 'and' after substrate; substitute a different word for 'Besides'
L#334: italicize Pseudomonas
L#329: change 'body' to 'bodies'
L#332: 'being the last one' is unclear in this context; chose different phrase to convey your meaning
L#364: delete 'whose' after exposure
L#366: delete 'literature', substitute 'recent report on H. marmoreus'
L#372: change 'bacteria' to 'bacterial
L#374: change 'the for' to 'therefore'
L#378: delete the B from BMetagenomic.
Fig. 4 legend needs to state whether the bars shown represent standard error or standard deviation. It also appears that day 12 was the only possibly statistically significant (p<0.05) increased biological efficiency application day for the 24 hr soaked MCCE treatment compared to the 1 hr. Some statement or graphical indication of statistical significance of the treatment diifferences would clarify this for readers. Such clarification would greatly value of the analyses presented.

Author Response
Dear reviewers and editor,
Thank you for your useful comments and suggestions of our manuscript.
We have modified the manuscript accordingly (highlighted in red text), and detailed corrections and comments are listed below point by point:
Reviewer #1: Corrections needed in the manuscript:
Line# 143: misspelled "Plhase". Okay, correct.
L#150: mushroom 'precocity' is unlikely to be understood by most readers even with the attempted escription. Is there another term that could be used. Okay, changed by precociousness as presented by Dias, E. S., Zied, D. C., & Rinker, D. L. (2013). Physiologic response of Agaricus subrufescens using different casing materials and practices applied in the cultivation of Agaricus bisporus. Fungal Biology, 117(7-8), 569-575.
L#216: change 'Over' to 'More". Okay, modified.
L#291: change 'on' to 'in'. Okay, modified.
L#296: change 'requiring' to 'require'. Okay, modified.
L#297: delete 'to'; substitute 'characterizing'. Okay, modified.
L#303: more complete than what?. Okay, modified.
L#316: italicize Bacillus. Okay, correct.
L#317: change 'fungi' to 'fungal'. Okay, changed.
L#322: delete 'it'. Okay, deleted.
L#323: change 'benefic' to 'beneficial'. Okay, changed.
L#325: ad 'and' after substrate; substitute a different word for 'Besides'. Okay, modified.
L#334: italicize Pseudomonas. Okay, correct.
L#329: change 'body' to 'bodies'. Okay, changed.
L#332: 'being the last one' is unclear in this context; chose different phrase to
convey your meaning. Okay, modified.
L#364: delete 'whose' after exposure. Okay, deleted.
L#366: delete 'literature', substitute 'recent report on H. marmoreus'. Okay, deleted.
L#372: change 'bacteria' to 'bacterial. Okay, changed.
L#374: change 'the for' to 'therefore'. Okay, changed.
L#378: delete the B from BMetagenomic. Okay, deleted.
Fig. 4 legend needs to state whether the bars shown represent standard error
or standard deviation. It also appears that day 12 was the only possibly
statistically significant (p<0.05) increased biological efficiency application day
for the 24 hr soaked MCCE treatment compared to the 1 hr. Some statement
or graphical indication of statistical significance of the treatment diifferences
would clarify this for readers. Such clarification would greatly improve the
understanding and value of the analyses presented. Okay, modified in the text not in the graphical area.

Reviewer 2 Report
Brief summary. Agaricus subrufescens are known worldwide due to its medicinal and nutritional properties. Metagenomics of extracts prepared from A. subrufescens compost in different soaking times consists in a powerful tool to elucidate the dynamics of the bacterial diversity and their interactions with the hyphae of basidiomycetes during its cultivation. The study is of practical importance due to the proven technology of preparation of the compost extract colonized with the A. subrufescens.
Manuscript is clear, relevant for the field and presented in a well-structured manner. The review is clear, comprehensive and of relevance to the field cited references are current. The manuscript is scientifically sound and is the experimental design appropriate to test the hypothesis. Figures/tables/images/schemes are appropriate and they properly show the data (easy to interpret and understand). The data are interpreted appropriately and consistently throughout the manuscript. The conclusions are consistent with the evidence and arguments presented.
Specific comments:
Unfortunately, in the «Materials and Methods» lacks data that are extremely important for understanding and reproducing the results of this research: pH compost, air and compost humidity, ventilation technology, carbon dioxide concentration. It is necessary to show in more detail the temperature regime (of compost and air). It is better to add detailed information in the form of a table. It must be in dynamics, at various stages of cultivation.
Author Response
Dear reviewers and editor,
Thank you for your useful comments and suggestions of our manuscript.
We have modified the manuscript accordingly (highlighted in red text), and detailed corrections and comments are listed below point by point:
Unfortunately, in the «Materials and Methods» lacks data that are extremely important for understanding and reproducing the results of this research: pH compost (was added to Table S1), air (was not analysed) and compost humidity (was added to Table S1), ventilation technology, carbon dioxide concentration (was added in the item 2.4. Casing Layer and Fruiting Induction). It is necessary to show in more detail the temperature regime (of compost and air). It is better to add detailed information in the form of a table. It must be in dynamics, at various stages of cultivation (was added to Figure S1).
